# Association of macular choroidal thickness with optical coherent tomography morphology in patients with idiopathic epiretinal membrane

I-Mo Fang[1,2]*, Li-Li Chen[1]

1 Department of Ophthalmology, Taipei City Hospital Zhongxiao Branch, Taipei, Taiwan, 2 Department of Ophthalmology, National Taiwan University Hospital, Taipei, Taiwan

* dah75@tpech.gov.tw

## Abstract

### Purpose

To compare macular choroidal thickness of idiopathic epiretinal membrane (ERM) and fellow eyes, and before and after vitrectomy in terms of the morphological features on spectral-domain optical coherence tomography (SD-OCT).

### Methods

Eighty-four patients with unilateral idiopathic ERM were involved. Patients were categorized into: Group 1, ERM without membrane contraction; Group 2, ERM with membrane contraction and retinal folding; and Group 3, ERM with membrane contraction and macular edema. Twenty-two patients received surgical removal of ERM. Choroidal thickness was compared between ERM and fellow eyes, and before and after treatment.

### Results

Mean choroidal thickness was significantly greater in ERM eyes than in fellow eyes (234.4 ± 22.5 vs 220.6 ± 20.8 μm; P<0.01). Group 1 (n = 20) showed no significant difference in choroidal thickness between ERM and fellow eyes. Eyes in Group 2 (n = 27) and Group 3 (n = 37) showed statistically significant differences in mean choroidal thickness between ERM and fellow eyes (229.6 ± 23.8 vs 220.8 ± 19.6 μm; 242.6 ± 27.8 vs 221.0 ± 21.8 μm, respectively; P<0.05). In Group 2 (n = 8) and Group 3 (n = 16), choroidal thickness in ERM eyes decreased significantly at 1 month and 6 months after surgery, compared with that before surgery (P<0.05 for all comparisons).

### Conclusions

Membrane contraction contributed to the increase in choroidal thickness in idiopathic ERM patients. This finding may help to elucidate the pathophysiologic features of idiopathic ERM as well as the response to treatment in these patients.

**Data Availability Statement:** All relevant data are within the manuscript.

**Funding:** The author(s) received no specific funding for this work

**Competing interests:** The authors have declared that no competing interests exist.

## Introduction

Epiretinal membrane (ERM) is a common macular disease that may induce various degrees of blurred vision and metamorphopsia [1–4]. At an early stage, ERM is characterized by an abnormal retinal light reflex on fundoscopy with mild retinal thickening and no distortion of the retinal surface [5, 6]. However, as the membrane becomes thicker, it may cause retinal folds, excursion or tortuosity of retinal vessels, resulting in increased retinal thickness, macular edema and disturbances in macular micro-circulation [7–9].

Spectral-domain optical coherent tomography (SD-OCT) facilitates the visualization of not only the retinal thickness but the ERM pathologic features and the associated retinal changes [10–12]. In addition, the recent development of enhanced depth imaging (EDI) has made choroidal examination with SD-OCT possible [13]. In the EDI technique, scan acquisition of the choroid-scleral interface is set up adjacent to the zero delay, where the sensitivity in SD-OCT images is highest. This allows better visualization of the choroids. Based on the anatomic structure of the macula and the vitreoretinal interface as shown on SD-OCT, Konidaris et al. classified ERM into two categories: without and with membrane contraction. The category of ERM with membrane contraction was further subdivided into that with retinal folding and that with edema. This classification enables a more accurate estimation of prognostic factors and will therefore help in accurately estimating postoperative results [14].

The foveola is an avascular zone which receives 100% of its blood supply from the choroid. Therefore, macular diseases may influence the choroidal blood flow and, in consequence, choroidal thickness. Several macular diseases, such as macular hole and age-related macular degeneration, have been reported to be associated with a change in choroidal thickness [15, 16]. Michalewska et al. first demonstrated decreased choroidal thickness after vitrectomy in patients, implying that the presence of ERM may affect choroidal thickness [17]. However, to our knowledge, no data have evaluated the influence of different morphological OCT findings on choroidal thickness in idiopathic ERM.

In this study, we postulated that macular traction and macular edema may disturb retinal circulation and then influence choroidal thickness in patients with idiopathic ERM. Therefore, we compared the macular choroidal thickness of ERM patients diagnosed by SD-OCT, with or without macular traction or macular edema. Furthermore, we evaluated the change in macular choroidal thickness before and after surgical removal of ERM to support our thesis.

## Methods

We retrospectively reviewed the medical records of consecutive patients with a diagnosis of unilateral idiopathic ERM who were examined at Taipei City Hospital, Zhongxiao Branch, from January 2, 2014 to June 30, 2017. Patients were excluded from this study who had eyes with secondary ERM (diabetic retinopathy, venous occlusion, retinal tear, retinal detachment, uveitis, trauma, etc.), eyes with myopia of more than 6 diopters, and eyes with other ocular pathologic features that could have interfered with functional results (such as glaucoma, visually significant cataract or age-related macular degeneration). In addition, patients with follow-up periods shorter than 6 months were also excluded from this study. The Institutional Review Board waived the need for patient consent for this retrospective study (Taipei City Hospital No: TCHIRB-10801013-E).

All patients underwent a full ophthalmic evaluation, including a slit-lamp examination and dilated ophthalmoscopy. Their spherical equivalent refractive error was measured with a TOP-CON KR-8100 autorefractor (Topcon, Tokyo, Japan), and then their cycloplegic refraction

was measured by Snellen chart under standard constant illumination. Snellen visual acuities were converted to logarithm of the minimal angle of resolution units for statistical analysis.

High resolution images of choroids were obtained using the vitreoretinal and chorioretinal settings of the EDI-OCT system (RTVue; Optovue, Inc., Fremont, CA, USA). Chorioretinal option provided a high resolution image of the choroid. Choroidal thickness was measured as previously reported [18]. Briefly, we obtained approximately 1024 A-scans on a 6 mm horizontal line passing through the centre of the fovea. By adjusting the contrast settings, we identified the image on which the choroid could be most clearly examined. Choroidal thickness was measured with horizontal B-scans on vertical lines running towards the chorioscleral junction, at the subfoveal region and 1.5 mm temporal and 1.5 mm nasal to the fovea. When the hyper-reflective chorioscleral junction could not be distinguished because of the shadowing caused by choroidal vessels, especially in the subfoveal region, the outer boundaries of the visible choroidal vessels were accepted as the chorioscleral junction. Each measurement was repeated three times, and the mean thickness level was calculated. The average from 3 choroidal thickness readings was recorded as the macular choroidal thickness.

To analyze central retinal thickness, an "MM5" grid-scanning mode was used. In this mode, 17 horizontal line scans and 17 vertical line scans are performed with each set of scans including 11 lines with a 5-mm scan length and 6 lines with a 3-mm scan length. After scanning in MM5 mode, the RTVue device automatically calculates the mean retinal thickness of the macular area. Central retinal thickness is the mean retinal thickness in the 1mm diameter circle centered on the fovea.

We classified patients into three groups: Group 1 represented patients with ERM without membrane contraction; Group 2 represented patients who had ERM with membrane contraction and associated retinal folding; and Group 3 represented patients who had ERM with membrane contraction and macular edema, according to the classification system proposed by Koniaris and colleagues [13].

### Pars plana vitrectomy

All 3-port 23-gauge pars plana vitrectomies were performed by one of the authors (IMF), using an Accurus machine (Alcon Surgical, Fort Worth, TX, USA). Vitreous body removal was followed by ERM removal. After the ILM was stained with 0.25% indocyanine green (ICG) solution, it was removed form an area within 2 or 3 disc diameters around the fovea, using microforceps. Postoperatively, all patients were advised to use topical antibiotic and steroid for 2 weeks (tobramycin and dexamethason)

### Statistical analysis

Statistical analysis was performed using SPSS version 18.0 software (SPSS, Inc., Chicago, IL, USA). The macular choroidal thickness of affected and fellow eyes was compared using the paired t-test. We used a repeated-measures general linear model to explore the significance of differences in measures before and after surgery. A P value of $<0.05$ indicated statistical significance.

### Results

There were 84 patients with unilateral ERM; their mean age was 64.9 ± 9.0 years (range, 42 to 84 years). Forty-seven (56.0%) of the patients were women. Forty-nine (58.3%) of the affected eyes were right eyes. Of the 84 ERM eyes, 20 (23.9%) were categorized into Group 1; 27 (32.1%) eyes were Group 2, and 37 (44.0%) were Group 3. Representative OCT images for every group are shown in Fig 1. The demographic data and clinical characteristics for every group are shown

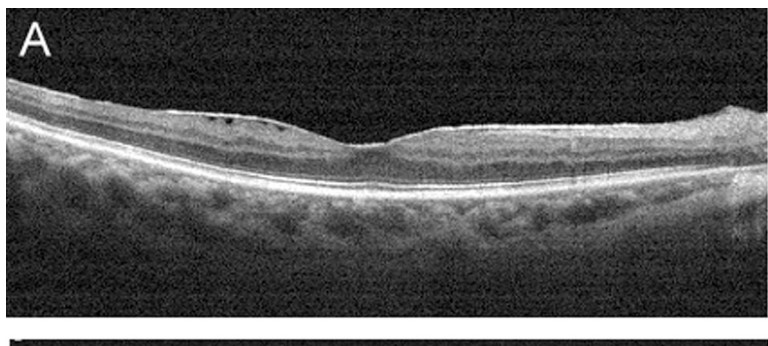

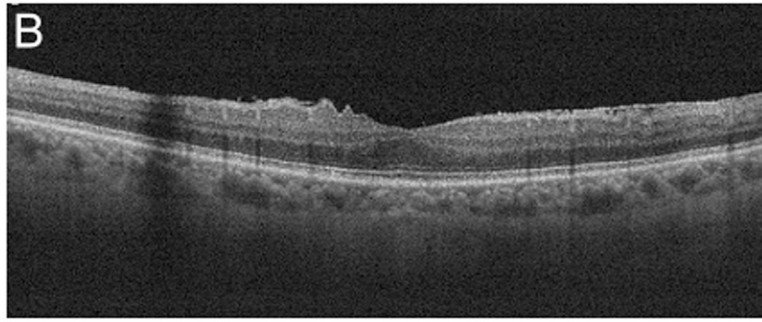

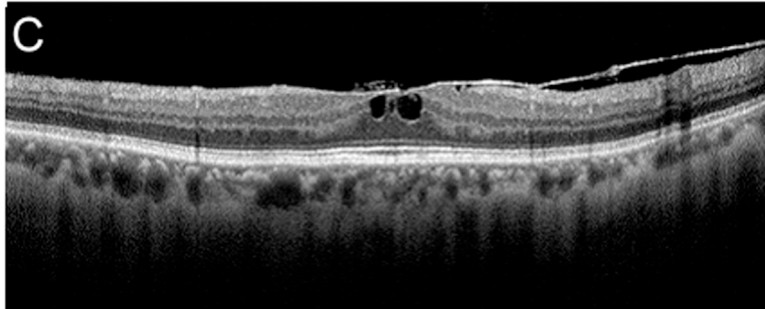

**Fig 1. Representative spectral-domain optical coherence tomography features of patients with epiretinal membrane (ERM).** Patients with ERM (A) without retinal contraction were classified as Group 1, (B) those with retinal contraction and folding as Group 2, and (C) those with retinal contraction and macular edema as Group 3.

in Table 1. The patients in Group 1 tended to have better visual acuity than those in Group 2 and Group 3 (P = 0.019). There was no statistically significant difference in age, percentage of women and hypertension among these three groups (P = 0.31, P = 0.18, P = 0.81, respectively). The ERM eyes and fellow eyes within each group had no statistically significant difference in mean axial length (P = 0.58 for group 1, P = 0.45 for group 2, P = 0.59 for group 3).

For all ERM patients studied, the mean ± standard deviation macular choroidal thickness was 234.4 ± 22.5 μm in ERM eyes and 220.6 ± 20.8 μm in healthy fellow eyes. Compared with healthy fellow eyes, ERM eyes showed significantly greater macular choroidal thickness (P < 0.01). The mean central retinal thickness of ERM eyes was 352.8 ± 89.1 μm. A strong correlation was found between macular choroidal thickness and central retinal thickness in ERM eyes (r = 0.35, P = 0.002). In Group 1, the mean macular choroidal thickness was 225.8 ± 22.1 μm in ERM eyes and 221.8 ± 21.1 μm in fellow eyes. The mean macular choroidal thickness did not differ statistically between the ERM and fellow eyes in Group 1 (P = 0.19). In Group 2, the mean macular choroidal thickness was 229.6 ± 23.8 μm in ERM eyes and 220.8 ± 19.6 μm in fellow eyes, a statistically significant difference (P = 0.01). In Group 3, the

**Table 1. Demographic and clinical characteristics of idiopathic epiretinal membrane and fellow eyes in different groups.**

| Patients features | ERM without contraction (Group 1) | | | ERM with retinal folding (Group 2) | | | ERM with retinal edema (Group 3) | | | P value |
|---|---|---|---|---|---|---|---|---|---|---|
| | ERM | Fellow eyes | p | ERM | Fellow eyes | p | ERM | Fellow eyes | p | |
| | N = 20 | | | N = 27 | | | N = 37 | | | |
| Age (yrs) mean ± SD | 62.4 ± 8.9 | | | 65.8 ± 7.6 | | | 64.8± 9.2 | | | 0.31 |
| No. of women (%) | 13 (65.0%) | | | 17 (65.4%) | | | 17 (45.9%) | | | 0.18 |
| Hypertension (%) | 3 (15.0%) | | | 5 (18.5%) | | | 6 (16.2%) | | | 0.81 |
| Axial length (mm) | 23.5± 0.8 | 23.4 ± 1.0 | 0.58 | 23.6 ± 1.0 | 23.4 ± 0.8 | 0.45 | 23.2 ± 1.0 | 23.3 ± 1.0 | 0.59 | 0.71 |
| BCVA, n (%) | | | | | | | | | | |
| logMAR | 0.29± 0.17 | 0.23± 0.13 | 0.02* | 0.58± 0.48 | 0.27± 0.19 | 0.001* | 0.56 ± 0.34 | 0.24 ± 0.18 | 0.001* | 0.019# |
| Snellen equlvalent | 0.52 ± 0.05 | 0.59± 0.04 | | 0.26± 0.17 | 0.56± 0.05 | | 0.28± 0.12 | 0.58 ± 0.05 | | |
| Macular choroidal thickness (µm) | 225.8± 22.1 | 221.8± 21.1 | 0.19 | 229.6± 23.8 | 220.8 ± 19.6 | 0.01* | 242.6± 27.8 | 221.0± 21.8 | 0.001* | 0.001# |
| Range (µm) | 167 to 274 | 169 to 258 | | 163.5 to 263 | 167 to 249 | | 175.5 to 283 | 158 to 255 | | |

yrs = years; SD = standard deviation; BCVA = best-corrected visual acuity; log MAR = logarithm of the minimal angle of resolution; p: comparison between ERM and fellow eyes

* <0.05; P value: comparison among ERM eyes among different groups,

# <0.05

mean macular choroidal thickness was 242.6 ± 27.8 µm in ERM eyes and 221.0 ± 21.8 µm in fellow eyes, also a statistically significant difference (P = 0.001)

Twenty-four patients received vitrectomy and surgical removal of ERM; in these patients, macular choroidal thickness was measured at 1 and 6 month after surgery. Eight of 27 patients (29.6%) in Group 2 and 16 of 37 patients (43.2%) in Group 3 received surgery. None of the patients in Group 1 received surgical intervention. The scatter plots of macular choroidal thickness, mean macular choroidal thickness and mean central retinal thickness in Group 2 and Group 3 patients measured before and at 1 and 6 months after surgery are shown in Fig 2. In Group 2, mean preoperative macular choroidal thickness was 239.6 ± 20.5 µm. After surgery, mean macular choroidal thickness decreased to 225.1 ± 15.7 µm at 1 month and 216.8 ± 11.2 µm at 6 months after surgery. The thicknesses at 1 and 6 months were significantly decreased, when compared with those preoperatively, for patients in Group 2 (P = 0.025 one month vs pre-operation; P = 0.004 six months vs pre-operation). Choroidal thickness in the fellow eyes did not differ significantly between measurement before and after surgery (P = 0.59 one month vs pre-operation; P = 0.61 six months vs pre-operation). Similarity, the mean central retinal thickness was 406.8 ± 63.5 µm in Group 2. After surgery, mean central retinal thickness decreased to 394.2 ± 59.0 µm at 1 month and 357.3 ± 34.4 µm at 6 months after surgery. The central retinal thickness was significantly decreased over time for patients in Group 2 (P = 0.001 one month vs pre-operation; P = 0.001 six months vs pre-operation).

In Group 3, the mean macular choroidal thickness was 245.9 ± 14.2 µm preoperatively, 232.3 ± 13.2 µm at 1 month and 225.5 ± 9.6 µm at 6 months after surgery. The choroidal thickness decreased significantly over time in Group 3 (P<0.001 one month vs pre-operation; P< 0.001 six months vs pre-operation). Similarity, in Group 3, the mean central retinal thickness was 432.4 ± 50.3 µm preoperatively, decreased to 413.1 ± 36.1 µm at 1 month and again to 361.3 ± 37.1 µm at 6 months after surgery, changes that also were statistically significant (P = 0.001 one month vs pre-operation; P< 0.001 six months vs pre-operation P<0.001). Fig 3 shows an example of changes in macular choroidal and central retinal thickness in Group 3 ERM patients before and at 1 month and 6 months after surgery.

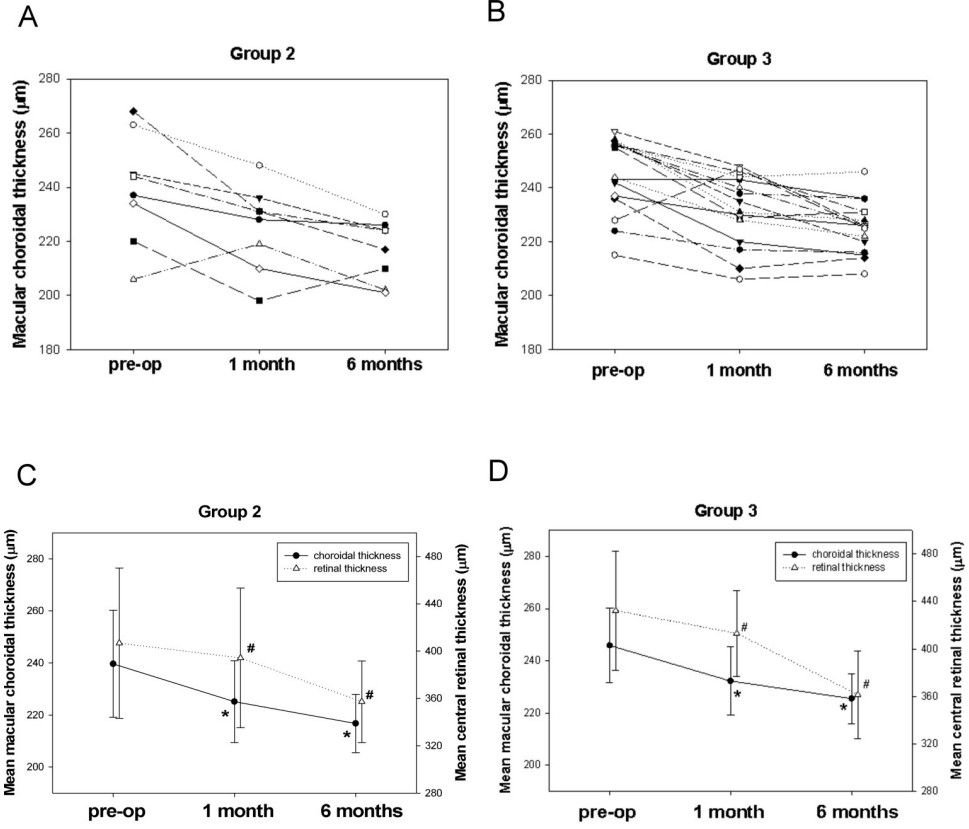

**Fig 2.** Scatter plots of macular choroidal thickness before and at 1 month and 6 months after surgery in (A) Group 2 and (B) Group 3 patients with epiretinal membrane. Mean choroidal thickness and mean central retinal thickness before and at 1 month and 6 months after surgery in (C) Group 2 and (D) Group 3 patients. All differences were statistically significant (*P<0.01 compared with preoperative mean macular choroidal thickness; #P<0.01 compared with preoperative mean central retinal thickness).

## Discussion

In this study, we demonstrated that the macular choroid thickness of ERM patients with membrane contraction in the macular area (that is, with either retinal folding or with macular edema) is significantly greater than that of fellow eyes. In contrast, ERM patients without membrane contraction showed no statistically significant difference in choroid thickness between the ERM eyes and healthy fellow eyes. Moreover, we found that the macular choroidal thickness decreased after surgical relief of macular membrane traction in ERM patients with retinal folding and in those with macular edema. Taken together, our results indicated that membrane contraction contributed to the increase in choroidal thickness in idiopathic ERM patients.

Several studies have reported that the tangential and vertical traction of ERM may cause retinal folds, macular edema, and vascular distortion, leading to altered hemodynamics in macular areas [19, 20]. Kadonosono et al. used fluorescein angiography to confirm that retinal capillary blood flow velocity is reduced in eyes with ERM [21, 22]. Yagi et al. showed that ERMs contributed to an increase in venous resistance and that surgical relief of retinal traction led to an improvement in the macular microcirculation [23, 24]. In our study, we observed that choroidal thickness was significantly increased in ERM patients with macular traction, but unchanged in patients without traction, when compared with the fellow eyes. It is possible

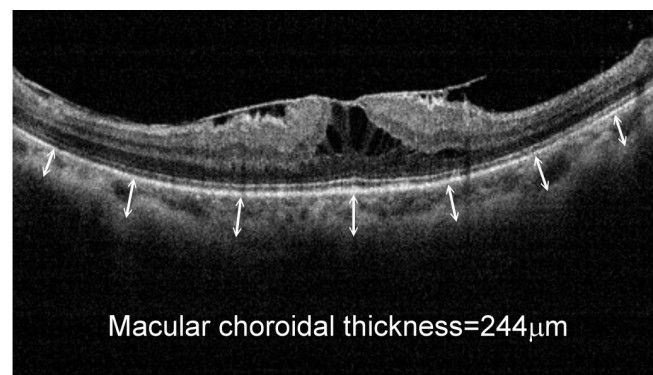

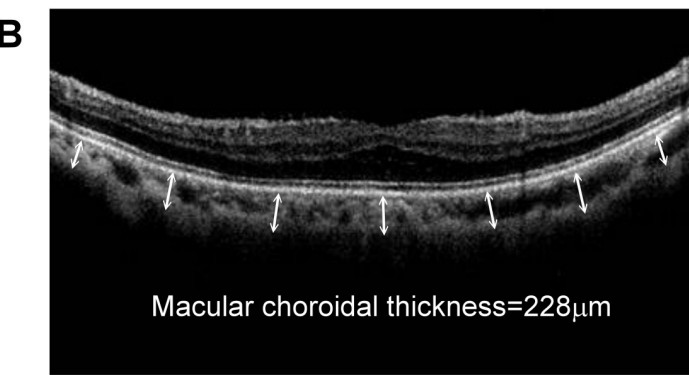

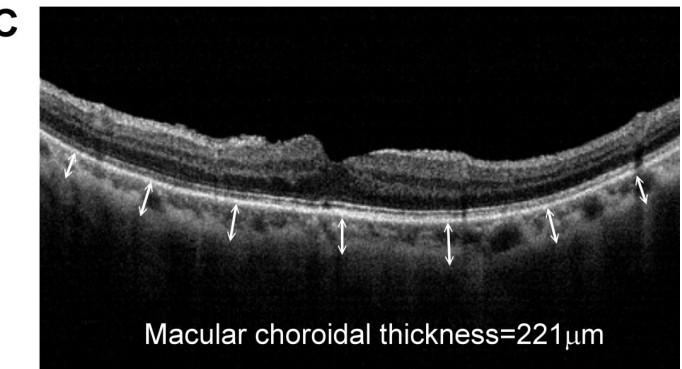

**Fig 3.** A representative optic coherent tomography of a 66-year-old female with idiopathic epiretinal membrane with retinal contraction and macular edema (Group 3) (A) before, and at (B) 1 month and (C) 6 months after surgery. Snellen visual acuity before surgery was 0.05, improved to 0.3 at 1 month postoperatively, and reached 0.7 at 6 months after surgery. Macular choroidal thickness decreased from 244 μm to 221 μm after 6 months. Similarity, central retinal thickness decreased from 492 μm to 248 μm after 6 months.

that the altered retinal blood flow caused by retinal traction may induce dilatation of the choroidal vessels to meet the demands for oxygen and nutrients in macular areas. Since the choroidal layer is rich in vessels, dilatation of the choroidal vessels may result in thickening of the choroid. Tsuiki et al. demonstrated that disturbance of retinal circulation in patients with central retinal vein occlusion (CRVO) is associated with an increase in choroidal thickness [25]. Our results showing that surgical removal of ERM, which resolved retinal traction, normalized not only retinal thickness but also choroidal thickness. These findings provide further evidence that macular traction contributes to the increase in macular choroidal thickness.

A previous study by Michalewsak et al. showed a similar result of decreased choroidal thickness at 3 months after surgical removal of ERM [17]. However, they did not find a significant difference in choroidal thickness between eyes with ERM and healthy fellow eyes, which differed from our observation. This discrepancy may be due to the different methods used to group patients in the two studies. In our study, patients were categorized by the presence of membrane contraction and macular edema, whereas the patients in the study by Michalewsak and colleagues were not classified.

This study had several limitations, including a small sample size and short-term follow up. Because the visual acuity of patients in Group 1 was generally good, no patients in this group received surgery. Therefore, we could not know the change in choroidal thickness in patients without macular traction. In addition, we used enhanced depth imaging (EDI) SD-OCT to measure choroidal thickness. By adjusting the zero delay position, or the position of the instrument relative to the patient's eye, it is possible for EDI imaging to image with the zero delay adjacent to the choroidal-scleral interface, such that sensitivity to the choroids is maximized. Branchini et al demonstrated good reproducibility between choroidal thickness measurements of images acquired with the Heidelberg Spectralis, Cirrus HD-OCT and RTvue in normal subjects [26]. Moreover, a previous study showed that age and axial length are critical in estimating choroidal thickness [27–29]. Although the groups in our study had no significant differences in age and axial length, other factors may have affected macular thickness. Further study with a large cohort of patients is required to reduce the potential bias and confirm results.

In conclusion, our results demonstrated that membrane contraction contributes to the increase in choroidal thickness in idiopathic ERM patients. Surgical removal of the epiretinal membrane not only relieved retinal traction and normalized retinal thickness, but also normalized choroidal thickness. These findings may help to elucidate the pathophysiologic features of idiopathic ERM as well as the response to treatment in these patients.

## Author Contributions

**Conceptualization:** I-Mo Fang.

**Data curation:** I-Mo Fang.

**Investigation:** I-Mo Fang.

**Methodology:** I-Mo Fang.

**Project administration:** I-Mo Fang.

**Resources:** Li-Li Chen.

**Supervision:** I-Mo Fang.

**Visualization:** Li-Li Chen.

**Writing – original draft:** I-Mo Fang.

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
