## [Decision Letter · Decision Letter 0]

15 Jun 2020

PONE-D-20-12131

Association of Macular Choroidal Thickness with Optical Coherent Tomography Morphology in Patients with Idiopathic Epiretinal Membrane

PLOS ONE

Dear Dr. I-Mo Fang,

Thank you for submitting your manuscript to PLOS ONE. After careful consideration, we feel that it has merit but does not fully meet PLOS ONE’s publication criteria as it currently stands. Therefore, we invite you to submit a revised version of the manuscript that addresses the points raised during the review process.

We look forward to receiving your revised manuscript.

Kind regards,

Alon Harris

Academic Editor

PLOS ONE

2. We noticed you have some minor occurrence of overlapping text with previous publications, which needs to be addressed:

https://doi.org/10.1016/j.ophtha.2011.07.002.

In your revision ensure you cite all your sources (including your own works), and quote or rephrase any duplicated text outside the methods section. Further consideration is dependent on these concerns being addressed.

3. Thank you for including your ethics statement:

'We have added the IRB approval number and institution in the manuscript'

a. Please amend your current ethics statement to include the full name of the ethics committee/institutional review board(s) that approved your specific study. and please confirm that your named institutional review board or ethics committee specifically approved this study.

Reviewers' comments:

Reviewer's Responses to Questions

**Comments to the Author**

1. Is the manuscript technically sound, and do the data support the conclusions?

Reviewer #1: No

Reviewer #2: Yes

2. Has the statistical analysis been performed appropriately and rigorously? 

Reviewer #1: I Don't Know

Reviewer #2: Yes

3. Have the authors made all data underlying the findings in their manuscript fully available?

Reviewer #1: No

Reviewer #2: Yes

4. Is the manuscript presented in an intelligible fashion and written in standard English?

Reviewer #1: No

Reviewer #2: Yes

5. Review Comments to the Author

Reviewer #1: Authors studies the epiretinal membrane and the association with choroidal thickness. They staged the ERM based on the features and compared before and after surgery to peel ERM and showed significant decrease on choroidal thickness. The findings are not novel, and the study had been repeated previously in multiple formats.

Major:

1. Anatomical speaking the fovea is 1.5mm in diameter while foveal avascular zone is only 0.5mm. Therefore, the notion that fovea receives 100% of its blood supply from choroidal is not correct.

2. The hypothesis that the fovea is vascularized by choroid therefore the epiretinal membrane should have choroidal changes is counterintuitive. The vice versa would be more relevant.

3. There are multiple reports on choroidal changes in ERM and novelty of the work is minimal. PMID: 31671195, PMID: 30467423, PMID: 26338821, PMID: 31139433, PMID: 28245442.

4. Only figure 1A was showing the EDI images and the rest are very poor quality.

Figures:

1. Figure 1. Poor scan quality with no visible choroidal structure in B and C, questioning the EDI mode of the scan.

2. Figure 3. Again, very poor contrast of the images makes it hard to evaluate choroidal thickness analysis.

Reviewer #2: It would be nice if the visual acuity before and after surgery would have reported and if the change of choirdal thickness correlates to the vision improvement

The study is well designed, the statistical work up is correct and literature review is good

6. PLOS authors have the option to publish the peer review history of their article (what does this mean?). If published, this will include your full peer review and any attached files.

Reviewer #1: No

Reviewer #2: No

---

## [Author Response · Author response to Decision Letter 0]

3 Aug 2020

July 5, 2020

Dear editor:

Thank you and the reviewers for the comprehensive review and valuable comments on our manuscript entitled “Association of Macular Choroidal Thickness with Optical Coherent Tomography Morphology in Patients with Idiopathic Epiretinal Membrane ‘’. We have revised the paper in accordance with the editor’s suggestions. In the following pages we will address each comment one by one. We appreciate the comments from the reviewers to improve the quality of our article. We hope our paper will be acceptable to be published in “PLOS ONE ’’.

Respectfully yours,

I-Mo Fang, M.D, Ph.D.

TEL: +886-2-27861288 ext.8275 

FAX: +886-2-27888492

E-mail: dah75@tpech.gov.tw

Reviewer 1:

Reviewer’s opinion 1#:

Anatomical speaking the fovea is 1.5mm in diameter while foveal avascular zone is only 0.5mm. Therefore, the notion that fovea receives 100% of its blood supply from choroidal is not correct.

Responses:

We totally agree and thanks to reviewer’s comment. The foveola is approximately 0.35 mm in diameter and lies in the center of the fovea and contains only cone cells. .In this region the cone receptors are found to be longer, slimmer and more densely packed than anywhere else in the retina, thus allowing that region to have the potential to have the highest visual acuity in the eye. We have changed the wording “ fovea” into “ foveola” in line 1, page 5.

Reviewer’s opinion 2#:

The hypothesis that the fovea is vascularized by choroid therefore the epiretinal membrane should have choroidal changes is counterintuitive. The vice versa would be more relevant.

Responses:

We very much agree with reviewer. Several studies have reported that the tangential and vertical traction of ERM may cause retinal folds, macular edema, and vascular distortion, leading to altered hemodynamics in macular areas. Membrane contraction of ERM patients contributes to the increase in choroidal thickness in idiopathic ERM patients. 

Reviewer’s opinion 3#:

There are multiple reports on choroidal changes in ERM and novelty of the work is minimal. PMID: 31671195, PMID: 30467423, PMID: 26338821, PMID: 31139433, PMID: 28245442.

Responses:

Actually, the changes of choroidal thickness or circulation is a hot toptic. Although there are various articles published on this topic, the focus of these articles is different. The title of PMID: 31671195 is “Macular Choroidal Thickness Changes in Development, Progression, and Spontaneous Resolution of Epiretinal Membrane”. This paper mainly focus on evaluating the association between macular choroidal thickness and the development, progression, and resolution of epiretinal membrane (ERM). The authors found that ERM with thining choroid is easy to progress, and ERM with thick choroid is easy to spontaneous resolve. The title of PMID: 30467423 is “Macular Microvasculature Features Before and After Vitrectomy in Idiopathic Macular Epiretinal Membrane: An OCT Angiography Analysis”. This study was to evaluate pre-operative and post-operative vascular parameters changes in ERM by optical coherence tomography angiography (OCTA). The title of PMID: 26338821 is Choroidal Thickness Change Following Vitrectomy in Idiopathic Epiretinal Membrane and Macular Hole. In this study, patient with macular hole were included, but our study were not. The title of PMID: 31139433 is “Mapping Diurnal Variations in Choroidal Sublayer Perfusion in Patients With Idiopathic Epiretinal Membrane: An Optical Coherence Tomography Angiography Study”. This study investigated diurnal changes in choroidal sublayer perfusion in eyes with ERM. In our study, we divided the patients into three morphological groups according to OCT findings. : Group 1, ERM without membrane contraction; Group 2, ERM with membrane contraction and retinal folding; and Group 3, ERM with membrane contraction and macular edema. The above studies did not have the same research as ours. Therefore, I believe that our research conclusion was still quite clinically valuable. Thanks reviewer for your reminder.

Reviewer’s opinion 4 #:

Only figure 1A was showing the EDI images and the rest are very poor quality.

Responses:

We thanks reviewer for the comment. Because the output graphic of OCT is a PDF file, the patient has cataract... and other factors, resulting in poor resolution of the images. We have try to revise the figure. 

Reviewer’s opinion 5 #:

Figures:

1. Figure 1. Poor scan quality with no visible choroidal structure in B and C, questioning the EDI mode of the scan.

2. Figure 3. Again, very poor contrast of the images makes it hard to evaluate choroidal thickness analysis.

Responses:

We have try to revise the figure 1 and 3. 

Reviewer 2:

Thanks reviewer for your encouragement.

---

## [Decision Letter · Decision Letter 1]

17 Sep 2020

Association of Macular Choroidal Thickness with Optical Coherent Tomography Morphology in Patients with Idiopathic Epiretinal Membrane

PONE-D-20-12131R1

Dear Dr. I-Mo Fang,

We’re pleased to inform you that your manuscript has been judged scientifically suitable for publication and will be formally accepted for publication once it meets all outstanding technical requirements.

Kind regards,

Alon Harris

Academic Editor

PLOS ONE

Additional Editor Comments (optional):

Reviewers' comments:

Reviewer's Responses to Questions

**Comments to the Author**

1. If the authors have adequately addressed your comments raised in a previous round of review and you feel that this manuscript is now acceptable for publication, you may indicate that here to bypass the “Comments to the Author” section, enter your conflict of interest statement in the “Confidential to Editor” section, and submit your "Accept" recommendation.

Reviewer #1: All comments have been addressed

Reviewer #2: All comments have been addressed

2. Is the manuscript technically sound, and do the data support the conclusions?

Reviewer #1: Yes

Reviewer #2: Yes

3. Has the statistical analysis been performed appropriately and rigorously? 

Reviewer #1: Yes

Reviewer #2: Yes

4. Have the authors made all data underlying the findings in their manuscript fully available?

Reviewer #1: Yes

Reviewer #2: (No Response)

5. Is the manuscript presented in an intelligible fashion and written in standard English?

Reviewer #1: Yes

Reviewer #2: Yes

6. Review Comments to the Author

Reviewer #1: The authors had made proper revision in text and figures. They addressed my concerns in the revision.

Reviewer #2: The authors in their response addressed explained and discussed well all the remarks of all reviewers

7. PLOS authors have the option to publish the peer review history of their article (what does this mean?). If published, this will include your full peer review and any attached files.

Reviewer #1: **Yes: **Amir Hajrasouliha

Reviewer #2: No

---

## [Editor Report · Acceptance letter]

18 Sep 2020

PONE-D-20-12131R1 

Association of Macular Choroidal Thickness with Optical Coherent Tomography Morphology in Patients with Idiopathic Epiretinal Membrane 

Dear Dr. Fang:

I'm pleased to inform you that your manuscript has been deemed suitable for publication in PLOS ONE. Congratulations! Your manuscript is now with our production department. 

Kind regards, 

on behalf of

Dr. Alon Harris 

Academic Editor

PLOS ONE